# GLGR: Question-aware Global-to-Local Graph Reasoning for Multi-party Dialogue Reading Comprehension

**Yanling Li[1], Bowei Zou[2], Yifan Fan[1], Xibo Li[1], Ai Ti Aw[2], Yu Hong[1]***

[1]School of Computer Science and Technology, Soochow University, Suzhou, China
[2]Institute for Infocomm Research, A*STAR, Singapore
{li4861988, yifanfannlp, cipolee20000210, tianxianer}@gmail.com
{zou_bowei, aaiti}@i2r.a-star.edu.sg

## Abstract

Graph reasoning contributes to the integration of discretely-distributed attentive information (clues) for Multi-party Dialogue Reading Comprehension (MDRC). This is attributed primarily to multi-hop reasoning over global conversational structures. However, existing approaches barely apply questions for anti-noise graph reasoning. More seriously, the local semantic structures in utterances are neglected, although they are beneficial for bridging across semantically-related clues. In this paper, we propose a question-aware global-to-local graph reasoning approach. It expands the canonical Interlocutor-Utterance graph by introducing a question node, enabling comprehensive global graph reasoning. More importantly, it constructs a semantic-role graph for each utterance, and accordingly performs local graph reasoning conditioned on the semantic relations. We design a two-stage encoder network to implement the progressive reasoning from the global graph to local. The experiments on the benchmark datasets Molweni and FriendsQA show that our approach yields significant improvements, compared to BERT and ELECTRA baselines. It achieves 73.6% and 77.2% F1-scores on Molweni and FriendsQA, respectively, outperforming state-of-the-art methods that employ different pretrained language models as backbones.

## 1 Introduction

MDRC is a special Machine Reading Comprehension (MRC) task. It involves answering questions conditioned on the utterances of multiple interlocutors (Yang and Choi, 2019; Li et al., 2020). MDRC presents unique challenges due to two aspects:

- MDRC relies heavily on multi-hop reasoning, where the necessary clues for reasoning are discretely distributed across utterances.

---
*Corresponding author.

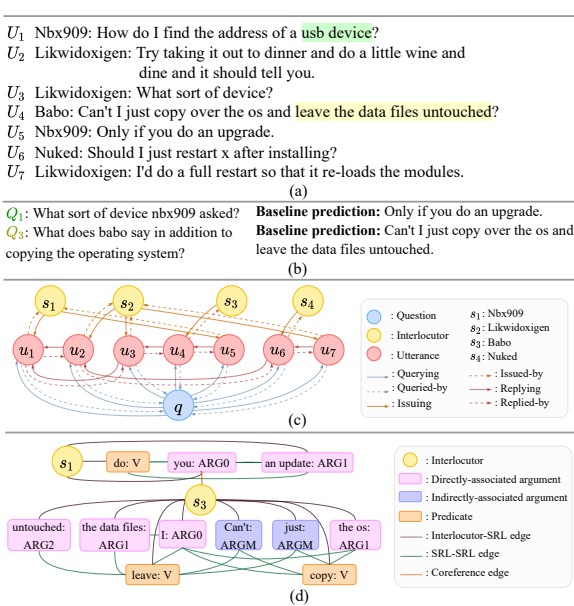

$U_1$ Nbx909: How do I find the address of a usb device?
$U_2$ Likwidoxigen: Try taking it out to dinner and do a little wine and dine and it should tell you.
$U_3$ Likwidoxigen: What sort of device?
$U_4$ Babo: Can't I just copy over the os and leave the data files untouched?
$U_5$ Nbx909: Only if you do an upgrade.
$U_6$ Nuked: Should I just restart x after installing?
$U_7$ Likwidoxigen: I'd do a full restart so that it re-loads the modules.

(a)

$Q_1$: What sort of device nbx909 asked? | **Baseline prediction:** Only if you do an upgrade.
$Q_3$: What does babo say in addition to copying the operating system? | **Baseline prediction:** Can't I just copy over the os and leave the data files untouched.

(b)

(c)

(d)

Figure 1: (a) A dialogue from Molweni (Li et al., 2020). (b) Two question-answering pairs of the dialogue. The correct answer to each question is highlighted in the dialogue with the same color as the question. (c) The QIUG of the dialogue, where edges between utterances indicate the discourse relationship between them (replying and replied-by). (d) The LSRG subgraph of $U_4$ and $U_5$ in the dialogue.

- Multi-hop reasoning suffers from discontinuous utterances and disordered conversations (see Figure 1-a,b).

Recently, a variety of graph-based multi-hop reasoning (abbr., graph reasoning) approaches have been proposed to tackle MDRC (Li et al., 2021; Ma et al., 2021, 2022). Graph reasoning is generally effective for bridging across the clues hidden in the discontinuous utterances, with less interference of redundant and distracting information occurring in the disordered conversations. The effectiveness is attributed primarily to the perception of global interactive relations of interlocutor-utterance graphs.

However, existing approaches encounter two bottlenecks. First, the question-disregarded graph construction methods (Li et al., 2021; Ma et al., 2021) fail to model the bi-directional interactions between

question and utterances. As a result, it is prone to involving question-unrelated information during reasoning. Second, the inner token-level semantic relations in every utterance are omitted, making it difficult to perceive the exact and unabridged clues occurring in the local contexts.

To address the issues, we propose a Global-to-Local Graph Reasoning approach (GLGR) with Pre-trained Language Models (PLMs) (Kenton and Toutanova, 2019; Clark et al., 2020) as backbones. It encodes two heterogeneous graphs, including Question-aware Interlocutor-Utterance Graph (QIUG) and Local Semantic Role Graph (LSRG). QIUG connects the question with all utterances in the canonical Interlocutor-Utterance graph (Figure 1-c). It depicts the global interactive relations. By contrast, LSRG interconnects fine-grained nodes (tokens, phrases and entities) in an utterance in terms of their semantic roles, where semantic role labeling (Shi and Lin, 2019) is used. It signals the local semantic relations. To enable connectivity between LSRGs of different utterances, we employ coreference resolution (Lee et al., 2018) and synonym identification to identify shareable nodes (Figure 1-d).

Methodologically, we develop a two-stage encoder network for progressive graph reasoning. It is conducted by successively encoding QIUG and LSRG, where attention modeling is used. The attentive information squeezed from QIUG and LSRG is respectively used to emphasize the global and local clues for answer prediction. Accordingly, the representation of the input is updated step-by-step during the progressive reasoning process. Residual network is used for information integration.

We carry out GLGR within an extractive MRC framework, where a pointer network (Vinyals et al., 2015) is used to extract answers from utterances. The experiments on Molweni (Li et al., 2020) and FriendsQA (Yang and Choi, 2019) demonstrate three contributions of this study, including:

- The separate use of QIUG and LSRG for graph reasoning yields substantial improvements, compared to PLM-based baselines.

- Global-to-local progressive reasoning on both graphs (i.e., GLGR) yields further improvements, allowing MDRC performance to reach 73.6% and 77.2% F1-scores, as well as 59.2% and 59.8% EM-scores.

- GLGR is stable. It obtains general improve-

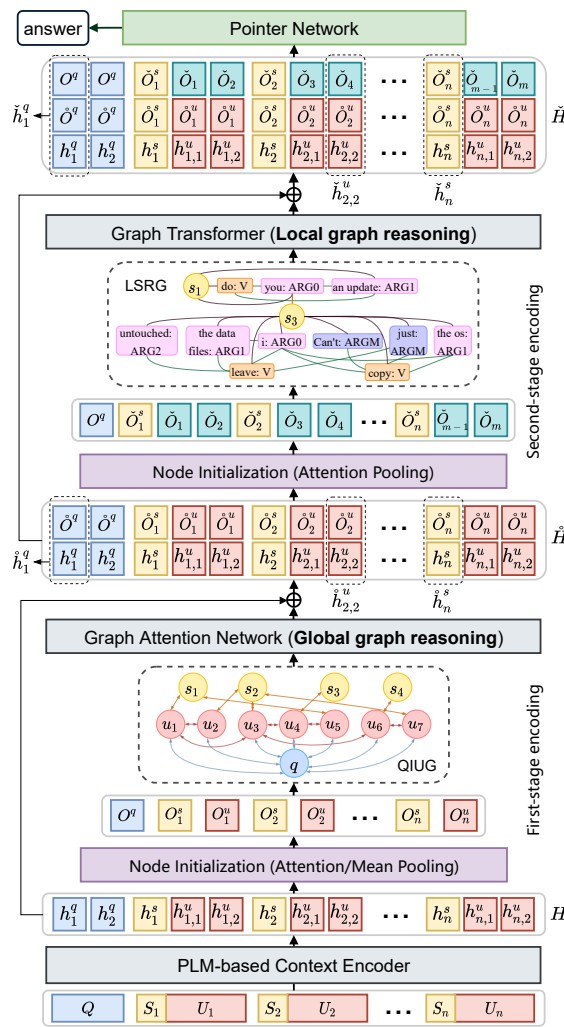

Figure 2: The main architecture of the two-stage encoder for Global-to-Local Graph Reasoning approach (GLGR).

ments when different PLMs are adopted as backbones (BERT and ELECTRA).

The rest of the paper is organized as follows. Section 2 presents the details of GLGR. We discuss the experimental results in Section 3, and overview the related work in Section 4. We conclude the paper in Section 5.

## 2 Approach

The overall architecture of GLGR-based MDRC model is shown in Figure 2. First of all, we utilize PLM to initialize the representation of the question, interlocutors and utterances. On this basis, the first-stage graph reasoning is conducted over QIUG, where Graph Attention network (GAT) (Veličković et al., 2018) is used for encoding. Subsequently, we carry out the second-stage graph reasoning over LSRG, where graph transformer layers (Zhu et al., 2019) are used for encoding. Finally, we concate-

nate the initial hidden states and their updated versions obtained by GLGR, and feed them into the pointer network for answer prediction.

In the following subsections, we detail computational details after presenting the task definition.

## 2.1 Task Definition

Formally, the task is defined as follows. Given a multi-party dialogue $D = \{U_1, U_2, ..., U_n\}$ with $n$ utterances and a question $Q$, MDRC aims to extract the answer $\mathcal{A}$ of $Q$ from $D$. When the question is unanswerable, $\mathcal{A}$ is assigned a tag "*Impossible*". Note that each utterance $U_i$ comprises the name of an interlocutor $S_i$ who issues the utterance, as well as the concrete content of the utterance.

## 2.2 Preliminary Representation

We follow Li and Zhao (2021)'s study to encode the question $Q$ and dialogue $D$ using PLM, so as to obtain token-level hidden states $H$ of all tokens in $Q$ and $D$. Specifically, we concatenate the question $Q$ and dialogue $D$ to form the input sequence $X$, and feed $X$ into PLM to compute $H$:

$$X = \{[CLS], Q, [SEP], U_1, [SEP], ..., U_n, [SEP]\}$$
$$H = PLM(X, \theta) \tag{1}$$

where, [CLS] and [SEP] denote special tokens. The hidden states $H \in \mathbb{R}^{l \times d}$ serve as the universal representation of $X$, where $l$ is the maximum length of $X$, and $d$ is the hidden size. The symbol $\theta$ denotes all the trainable parameters of PLM. In our experiments, we consider three different PLMs as backbones in total, including BERT-base-uncased ($\text{BERT}_{base}$), BERT-large-uncased ($\text{BERT}_{large}$) (Kenton and Toutanova, 2019) and ELECTRA (Clark et al., 2020).

To facilitate understanding, we clearly define different levels of hidden states as follows:

- $H$ refers to the hidden states of all tokens in $X$, i.e., the universal representation of $X$. $h$ is the hidden state of a token $x$ ($x \in X$).

- $O$ is the hidden state of a specific node, known as node representation. Specifically, $O^q$, $O^s$ and $O^u$ denote the representations of question node, interlocutor node and utterance node.

## 2.3 Global-to-Local Graph Reasoning

We carry out Global-to-Local Graph Reasoning (GLGR) to update the hidden states $H$ of the input sequence $X$. GLGR is fulfilled by two-stage progressive encoding over two heterogeneous graphs, including QIUG and LSRG.

### 2.3.1 Global Graph Reasoning on QIUG

**QIUG**– QIUG is an expanded version of the canonical interlocutor-utterance graph (Li et al., 2021; Ma et al., 2021) due to the involvement of question-oriented relations, as shown in Figure 1-(c).

Specifically, QIUG comprises one question node, $N_u$ utterance nodes and $N_s$ interlocutor nodes. We connect the nodes using the following scheme:

- Each question node is linked to all utterance nodes. Bidirectional connection is used, in the directions of "querying" and "queried-by".

- Each interlocutor node is connected to all the utterance nodes she/he issued. Bidirectional connection is used, in the directions of "issuing" and "issued-by".

- Utterance nodes are connected to each other in terms of Conversational Discourse Structures (CDS) (Liu and Chen, 2021; Yu et al., 2022). CDS is publicly available in Molweni (Li et al., 2020), though undisclosed in FriendsQA (Yang and Choi, 2019). Therefore, we apply Liu and Chen (2021)'s open-source CDS parser to tread with FriendsQA. Bidirectional connection is used, i.e., in the directions of "replying" and "replied-by".

Consequently, QIUG contains six types of interactive relations $T=\{$querying, queried-by, issuing, issued-by, replying, replied-by$\}$. Each directed edge in QIUG solely signals one type of relation.

**Node Representation**– For an interlocutor node $S_i$, we consider the tokens of her/his name and look up their hidden states in the universal representation $H$. We aggregate the hidden states by mean pooling (Gholamalinezhad and Khosravi, 2020). The resultant embedding $O_i^s \in \mathbb{R}^{1 \times d}$ is used as the node representation of $S_i$.

For an utterance node $U_i$, we aggregate the hidden states of all tokens in $U_i$. Attention pooling (Santos et al., 2016) is used for aggregation. The resultant embedding $O_i^u \in \mathbb{R}^{1 \times d}$ is used as the node representation of $U_i$. We obtain the representation $O^q$ of the question node $Q$ in a similar way.

**Multi-hop Reasoning**– Multi-hop reasoning is used to discover and package co-attentive information across nodes and along edges. Methodologically, it updates the hidden states of all tokens in a node using the attentive information $\mathring{O}$ in the neighboring nodes. Formally, the hidden state of each

token is updated by: $\mathring{h}=[h;\mathring{O}]$, i.e., concatenating $h \in \mathbb{R}^{1 \times d}$ and $\mathring{O} \in \mathbb{R}^{1 \times d}$.

We utilize $L_1$-layer GAT (Veličković et al., 2018) networks to compute $\mathring{O}$, where attention-weighted information fusion is used:

$$\mathring{O}_i^{(l+1)} = \sum_{O_j \in E_i} \alpha_{i,j}^{(l)} \mathring{W}^{(l)} O_j^{(l)} \quad (2)$$

where, $E_i$ comprises a set of neighboring nodes of the $i$-th node $O_i$. $\mathring{W}$ is a trainable scalar parameter, and the superscript $(l)$ signals the computation at the $l$-the layer of GAT. Besides, $\alpha_{i,j}$ is a node-level attention score of $O_i$ ($O_i \notin E_i$) relative to $O_j$ ($O_j \in E_i$). The resultant $\mathring{O}_i^{L_1}$ is used to update the hidden states of all tokens in the $i$-th node.

**Divide-and-conquer Attention Modeling**– Different interactive relations have distinctive impacts on attention modeling. For example, the "queried-by" relation (i.e., an edge directed from utterance to question) most likely portends the payment of more attention to the possible answer in the utterance. By contrast, the "replying" and "replied-by" relations (i.e., edges between utterance nodes) induce the payment of more attention to the complementary clues in the utterances. In order to distinguish between the impacts, we separately compute node-level attention scores $\alpha$ for different types of edges in QIUG. Given two nodes $O_i$ and $O_j$ with a $t$-type edge, the attention score $\alpha_{i,j}$ is computed as:

$$\alpha_{i,j}^{(l)} = \frac{\exp(f([O_i^{(l)}; O_j^{(l)}], \dot{W}^{(l)}))}{\sum_{O_k \in E_{i,t}} \exp(f([O_i^{(l)}; O_k^{(l)}], \dot{W}^{(l)}))} \quad (3)$$

where $[;]$ is the concatenation operation, while $f(\cdot)$ denotes the LeakyRelu (Maas et al., 2013) activation function. $E_{i,t}$ is the set of all neighboring nodes that are connected to $O_i$ with a $t$-type edge. $\dot{W}_t^{(l)} \in \mathbb{R}^{2d \times 1}$ is a trainable parameter.

**Prediction and Training**– Using multi-hop reasoning, we reproduce the universal representation: $\mathring{H}=[H;\mathring{O}_{all}^{L_1}]$. We feed $\mathring{H}$ into a two-layer pointer network to predict the answer, determining the *start* and *end* positions of the answer in $X$. Note that the hidden state of [CLS] in $\mathring{H}$ is used to predict the "*Impossible*" tag, i.e., a tag signaling unanswerable question. During training, we use the cross-entropy loss function to optimize the model.

### 2.3.2 Local Graph Reasoning on LSRG

Global graph reasoning is grounded on the global relations among question, interlocutor and utterance nodes, as well as their indecomposable node representations. It barely uses the inner token-level semantic relations in every utterance for multi-hop

reasoning. However, such local semantic correlations actually contribute to the reasoning process, such as the *predicate-time* and *predicate-negation* relations, as well as coreferential relations. Therefore, we construct the semantic-role graph LSRG, and use it to strengthen local graph reasoning.

**LSRG**– LSRG is an undirected graph which comprises semantic-role subgraphs of all utterances in $D$. To obtain the subgraph of an utterance, we leverage Allennlp-SRL parser (Shi and Lin, 2019) to extract the predicate-argument structures in the utterance, and regard predicates and arguments as the fine-grained nodes. Each predicate node is connected to the associated argument nodes with undirected role-specific edges (e.g., "ARG1-V"). Both the directly-associated arguments and indirectly-associated are considered for constructing the subgraph, as shown in Figure 1-(d).

Given the semantic-role subgraphs of all utterances, we form LSRG using the following scheme:

- We combine the subgraphs containing similar fine-grained nodes. It is fulfilled by connecting the similar nodes. A pair of nodes is determined to be similar if their inner tokens have an overlap rate more than 0.5.

- Interlocutor name is regarded as a special fine-grained node. We connect it to the fine-grained nodes in the utterances she/he issued.

- We combine subgraphs containing coreferential nodes. Lee et al. (2018)'s coreference resolution toolkit[1] is used.

**Fine-grained Node Representation**– The fine-grained nodes generally contain a varied number of tokens (e.g., "*can not*" and "*the data files*"). To obtain identically-sized representations of them, we aggregate the hidden states of all tokens in each fine-grained node. Attention pooling (Santos et al., 2016) is used for aggregation.

In our experiments, there are two kinds of token-level hidden states considered for fine-grained node representation and reasoning on LSRG, including the initial case $h$ obtained by PLM, as well as the refined case $\mathring{h}$ ($\mathring{h} \in \mathring{H}$) by global graph reasoning. When $h$ is used, we perform local graph reasoning independently, without the collaboration of global graph reasoning. It is carried out for MDRC in an ablation study. When $\mathring{h}$ is used, we perform global-to-local graph reasoning. It is conducted in

---

[1]https://demo.allennlp.org/coreference-resolution

the comparison test. We concentrate on $\mathring{h}$ in the following discussion.

**Multi-hop Reasoning on LSRG**– It updates hidden states of all tokens in each fine-grained node $O_i$, where the attentive information $\check{O}$ of its neighboring nodes in LSRG is used for updating. Formally, the hidden state of each token is updated by: $\check{h}=[\mathring{h};\check{O}]$. We use a $L_2$-layer graph transformer (Zhu et al., 2019) to compute $\check{O} \in \mathbb{R}^{1 \times d}$ as follows:

$$\check{O}_i^{(l+1)} = \sum_{O_j \in E_i} \beta_{ij}^{(l)} \left( O_j^{(l)} \check{W}_o^{(l)} + r_{ij}^{(l)} \check{W}_r^{(l)} \right) \quad (4)$$

where, $E_i$ is a set of neighboring fine-grained nodes of the $i$-th node $O_i$ in LSRG. $\check{W}_o \in \mathbb{R}^{d \times d}$ and $\check{W}_r \in \mathbb{R}^{d \times d}$ are trainable parameters. In addition, $r_{i,j} \in \mathbb{R}^{1 \times d}$ is the learnable embedding of the role-specific edge between $O_i$ and its $j$-th neighboring node $O_j$. Before training, the edges holding the same semantic-role relation are uniformly assigned a randomly-initialized embedding. Besides, $\beta_{i,j}$ is a scalar, denoting the attention score between $O_i$ and $O_j$. It is computed as follows:

$$\beta_{ij}^{(l)} = \frac{exp\left(e_{ij}^{(l)}\right)}{\sum_{i=1}^{M} exp\left(e_{ik}^{(l)}\right)} \quad (5)$$

$$e_{ij}^{(l)} = O_i^{(l)} \dot{W}_e^{(l)} \left( O_j^{(l)} \dot{W}_o^{(l)} + r_{ij}^{(l)} \dot{W}_r^{(l)} \right)^T / \sqrt{d} \quad (6)$$

where, $\dot{W}_e^{(l)}$, $\dot{W}_o^{(l)}$ and $\dot{W}_r^{(l)}$ are trainable parameters of dimensions $\mathbb{R}^{d \times d}$. $M$ denotes the number of neighboring fine-grained nodes in $E_i$.

**Question-aware Reasoning**– Obviously, the LSRG-based attentive information $\check{O}_i$ is independent of the question. To fulfill question-aware reasoning, we impose the question-oriented attention upon $\check{O}_i$. Formally, it is updated by: $\check{O}_i \Leftarrow \gamma_i \cdot \check{O}_i$, where the attention score $\gamma_i$ is computed as follows:

$$\gamma_i = \text{Sigmoid}\left(O^q W_q O_i / \sqrt{d}\right) \quad (7)$$

where, $O^q$ is the representation of the question node, and $W_q \in \mathbb{R}^{d \times d}$ is a trainable parameter.

**Prediction and Training**– To fulfill global-to-local reasoning (GLGR), we reproduce the universal representation: $\check{H}=[\check{H};\check{O}_{all}^{L2}]$. It is actually equivalent to the operation of concatenating the initial representation $H$, QIUG-based co-attentive information $\mathring{O}_{all}^{L1}$ and LSRG-based co-attentive information $\check{O}_{all}^{L2}$, i.e., $\check{H}=[H;\mathring{O}_{all}^{L1};\check{O}_{all}^{L2}]$.

In the scenario of independent local graph reasoning, the QIUG-based co-attentive information is omitted. Accordingly, the universal representation is calculated as: $\check{H}=[H;\check{O}_{all}^{L2}]$, where $\check{O}_{all}^{L2}$ is computed using $h$ instead of $\mathring{h}$.

| Datasets | Molweni | | | FriendsQA | | |
|---|---|---|---|---|---|---|
| | Train | Dev | Test | Train | Dev | Test |
| Questions | 24,682 | 2,513 | 2,871 | 8,535 | 1,010 | 1,065 |
| Utterances | 77,374 | 7,823 | 2,513 | 21,607 | 2,847 | 2,336 |
| Dialogues | 8,771 | 883 | 100 | 977 | 122 | 123 |

Table 1: Statistics in Molweni and FriendsQA.

We feed $\check{H}$ into the two-layer pointer network for predicting answers. Cross-entropy loss is used to optimize the model during training.

## 3 Experimentation

### 3.1 Datasets and Evaluation

We experiment on two benchmark datasets, including Molweni (Li et al., 2020) and FriendsQA (Yang and Choi, 2019). Molweni is an MDRC dataset manufactured using Ubuntu Chat Corpus (Lowe et al., 2015). The dialogues in Molweni are accompanied with ground-truth CDS, as well as either answerable or unanswerable questions. FriendsQA is another MDRC dataset whose dialogues are excerpted from the TV show *Friends*. It is characterized by colloquial conversations. CDS is undisclosed in FriendsQA. We use Liu and Chen (2021)'s CDS parser to pretreat the dataset.

We follow the common practice (Li et al., 2020; Yang and Choi, 2019) to split Molweni and FriendsQA into the training, validation and test sets. The data statistics about the sets are shown in Table 1. We use F1-score and EM-score (Li and Zhao, 2021; Ma et al., 2021) as the evaluation metrics.

### 3.2 Backbone and Hyperparameter Settings

To verify the stability of GLGR, we construct three GLGR-based models using different PLMs as backbones, including BERT-base-uncased ($\text{BERT}_{base}$), BERT-large-uncased ($\text{BERT}_{large}$) (Kenton and Toutanova, 2019) and ELECTRA (Clark et al., 2020). The hyperparameters are set as follows.

All the models are implemented using Transformers Library (Li and Choi, 2020). AdamW optimizer (Loshchilov and Hutter, 2018) is used for training. Towards the experiments on Molweni, we set the batch size to 16 for BERT and 12 for ELECTRA. The learning rates are set to 1.8e-5, 5.2e-5 and 1e-5 for $\text{BERT}_{base}$, $\text{BERT}_{large}$ and ELECTRA, respectively. For FriendsQA, we set the batch sizes to 16, 8 and 12 for $\text{BERT}_{base}$, $\text{BERT}_{large}$ and ELECTRA, respectively. The learning rates are set to 1.8e-5 for $\text{BERT}_{base}$ and 1e-5 for both $\text{BERT}_{large}$ and ELECTRA backbone. In addition,

| Model | Molweni | | FriendsQA | |
|---|---|---|---|---|
| | EM | F1 | EM | F1 |
| *BERT*$_{base}$ (Kenton and Toutanova, 2019) | | | | |
| Baseline | 47.6 | 61.6 | 42.7 | 60.0 |
| DADgraph (Li et al., 2021) | 46.5 | 61.5 | - | - |
| ULM-UOP (Li and Choi, 2020) | - | - | 46.8 | 63.1 |
| BiDeN (Li et al., 2022) | 48.1 | 63.2 | - | - |
| SKIDB (Li and Zhao, 2021) | 49.2 | 64.0 | 46.9 | 63.9 |
| ESA (Ma et al., 2021) | **49.7** | 64.4 | **47.0** | 63.0 |
| GLGR$^{\dagger}$ (Ours) | 48.2 | 64.4 | 47.0 | 64.3 |
| *BERT*$_{large}$ (Kenton and Toutanova, 2019) | | | | |
| Baseline | 50.4 | 65.8 | 46.0 | 63.3 |
| ESA (Ma et al., 2021) | 52.9 | 66.9 | 49.0 | 64.0 |
| GLGR$^{\dagger}$ (Ours) | **53.7** | **67.5** | **49.8** | **67.1** |
| *ELECTRA* (Clark et al., 2020) | | | | |
| Baseline | 56.8 | 70.6 | 57.0 | 74.8 |
| SKIDB (Li and Zhao, 2021) | 58.0 | 72.9 | 55.8 | 72.3 |
| ESA (Ma et al., 2021) | 58.6 | 72.2 | 58.7 | 75.4 |
| GLGR$^{\dagger}$ (Ours) | **59.2** | **73.6** | **59.8** | **77.2** |

Table 2: Results on the test sets of Molweni and FriendsQA. The superscript † denotes a statistically significant improvement of F1-score ($p < 0.05$) compared to the baseline.

The numbers of network layers of GAT and graph transformer are set in the same way: $L_1=L_2=2$.

### 3.3 Compared MDRC Models

Following Ma et al. (2021)'s study, we consider the standard span-based MRC model (Kenton and Toutanova, 2019) as the baseline. We compare with a variety of state-of-the-art MDRC models:

- **DADgraph** (Li et al., 2021) constructs a CDS-based dialogue graph. It enables the graph reasoning over conversational dependency features and interlocutor nodes. Graph Convolutional Network (GCN) is used for reasoning.

- **ULM-UOP** (Li and Choi, 2020) fine-tunes BERT on a larger number of FriendsQA transcripts (known as Character Mining dataset (Yang and Choi, 2019)) before task-specific training for MDRC. Two self-supervised tasks are used for fine-tuning, including utterance-oriented masked language modeling and utterance order prediction. In addition, BERT is trained to predict both answers and sources (i.e., IDs of utterances containing answers).

- **SKIDB** (Li and Zhao, 2021) uses a multi-task learning strategy to enhance MDRC model. Self-supervised interlocutor prediction and key-utterance prediction tasks are used within the multi-task framework.

- **ESA** (Ma et al., 2021) uses GCN to encode the interlocutor graph and CDS-based utterance graph. In addition, it is equipped with a speaker masking module, which is able to highlight co-attentive information within utterances of the same interlocutor, as well as that among different interlocutors.

- **BiDeN** (Li et al., 2022) incorporates latent information of utterances using different temporal structures (e.g., future-to-current, current-to-current and past-to-current interactions).

### 3.4 Main Results

Table 2 shows the test results of our GLGR models and the compared models, where different backbones (BERT$_{base}$, BERT$_{large}$ and ELECTRA) are considered for the general comparison purpose.

It can be observed that our GLGR model yields significant improvements, compared to the PLM baselines. The most significant performance gains are 4.3% F1-score and 4.3% EM-score, which are obtained on FriendsQA compared to the lite BERT$_{base}$ (110M parameters). When compared to the larger BERT$_{large}$ (340M parameters) and ELECTRA (335M parameters), GLGR is able to achieve the improvements of no less than 1.7% F1-score and 2.4% EM-score. In addition, GLGR outperforms most of the state-of-the-art MDRC models. The only exception is that GLGR obtains a comparable performance relative to ESA when BERT$_{base}$ is used as the backbone. By contrast, GLGR substantially outperforms ESA when BERT$_{large}$ and ELECTRA are used.

The test results reveal distinctive advantages of the state-of-the-art MDRC models. **DADgraph** is a vest-pocket model due to the involvement of a sole interlocutor-aware CDS-based graph. It offers the basic performance of graph reasoning for MDRC. **ESA** is grounded on multiple graphs, and it separately analyzes co-attentive information for subdivided groups of utterances. Multi-graph reasoning and coarse-to-fine attention perception allow ESA to be a competitive MDRC model. By contrast, **ULM-UOP** doesn't rely heavily on conversational structures. Instead, it leverages larger dataset and diverse tasks for fine-tuning, and thus enhances the ability of BERT in understanding domain-specific languages at the level of semantics. It can be observed that ULM-UOP achieves similar performance compared to ESA. **SKIDB** successfully leverages multi-task learning, and it applies interesting and effective self-supervised learning tasks. Similarly, it enhances PLMs in encoding domain-specific languages, which is not limited to

| Model | Molweni | | FriendsQA | |
|---|---|---|---|---|
| | EM | F1 | EM | F1 |
| GLGR | **59.2** | **73.6** | **59.8** | **77.2** |
| w/o QIUG | 58.1 | 72.3 | 58.3 | 76.4 |
| w/o LSRG | 57.7 | 72.6 | 58.9 | 76.0 |
| w/o QIUG& LSRG (baseline) | 56.8 | 70.6 | 57.0 | 74.8 |

Table 3: Graph ablation study, where ELECTRA (Clark et al., 2020) is considered as the backbone in this study.

| Model | Molweni | | FriendsQA | |
|---|---|---|---|---|
| | EM | F1 | EM | F1 |
| QIUG | **57.7** | **72.6** | **58.9** | **76.0** |
| w/o interloctuor | 57.0 | 71.3 | 57.7 | 75.6 |
| w/o question | 57.2 | 71.0 | 57.5 | 75.6 |
| w/o utterance structure | 57.3 | 71.1 | 57.8 | 75.2 |
| LSRG | **58.1** | **72.3** | **58.3** | **76.4** |
| w/o coreference | 57.2 | 71.5 | 57.8 | 75.6 |
| w/o semantic role relation | 56.6 | 71.5 | 57.8 | 75.3 |
| w/o question-aware reasoning | 56.8 | 71.8 | 58.1 | 75.4 |

Table 4: Relation ablation study, whre ELECTRA (Clark et al., 2020) is considered as the backbone in this study.

BERT. It can be found that SKIDB obtains comparable performance on Molweni compared to ESA.

Our GLGR model combines the above advantages by external conversational structure analysis and internal semantic role analysis. On this basis, GLGR integrates global co-attentive information with local for graph reasoning. It can be observed that GLGR shows superior performance, although it is trained without using additional data.

## 3.5 Ablation Study

We conduct the ablation study from two aspects. First of all, we progressively ablate global and local graph reasoning, where QIUG and LSRG are omitted accordingly. Second, we respectively ablate different classes of edges from the two graphs, i.e., disabling the structural factors during multi-hop reasoning. We refer the former to "Graph ablation" while the latter the "Relation ablation".

**Graph ablation**– The negative effects of graph ablation is shown in Table 3. It can be easily found that similar performance degradation occurs when QIUG and LSRG are independently pruned. This implies that local reasoning on LSRG is effectively equivalent to global reasoning on QIUG, to some extent. When graph reasoning is thoroughly disabled (i.e., pruning both QIUG and LSRG), the performance degrades severely.

**Relation ablation**– The negative effects of relation ablation is shown in Table 4. For QIUG, we condense the graph structure by respectively disabling interlocutor, question and utterance nodes. It

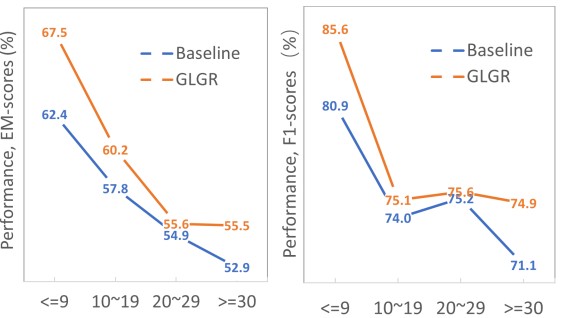

Figure 3: Performance of GLGR (ELECTRA) and baseline on the FriendsQA with different utterance numbers.

can be found that the performance degradation rates in the three ablation scenarios are similar. This demonstrates that all the conversational structural factors are crucial for multi-hop reasoning. For LSRG, we implement relation ablation by unbinding the co-referential fine-grained nodes, omitting semantic-role relations, and removing question-aware reasoning, respectively. The way to omit semantic-role relations is accomplished by full connection. It can be observed that ablating semantic-role relations causes relatively larger performance degradation rates.

## 3.6 The Impact of Utterance Number

We follow the common practice (Li and Zhao, 2021; Li et al., 2022) to verify the impacts of utterance numbers. The FriendsQA test set is used in the experiments. It is divided into four subsets, including the subsets of dialogues containing 1) no more than 9 utterances, 2) 10∼19 utterances, 3) 20∼29 utterances and 4) no less than 30 utterances. The GLGR is re-evaluated over the subsets. We illustrate the performance in Figure 3.

It can be found that the performance decreases for the dialogues containing a larger number of utterances, no matter whether the baseline or GLGR is used. In other words, both the models are distracted by plenty of noises in these cases. Nevertheless, we observe that GLGR is able to stem the tide of performance degradation to some extent. Therefore, we suggest that GLGR contributes to the anti-noise multi-hop reasoning, although it fails to solve the problem completely.

## 3.7 Reliability of Two-stage Reasoning

There are two alternative versions of GLGR: one changes the reason sequence of QIUG and LSRG (local-to-global graph reasoning). The other performs the single-phase reasoning over a holistic graph that interconnects QIUG and LSRG. In this

| Model | Molweni | | FriendsQA | |
|---|---|---|---|---|
| | EM | F1 | EM | F1 |
| GLGR (Two-stage) | **59.2** | **73.6** | **59.8** | **77.2** |
| GLGR (Two-stage rev.) | 58.3 | 72.7 | 59.1 | 76.2 |
| GLGR (Single-phase) | 58.0 | 72.0 | 58.5 | 75.4 |

Table 5: Comparing single-phase GLGR and two two-stage GLGRs, here "rev." indicates the local-to-global reasoning version.

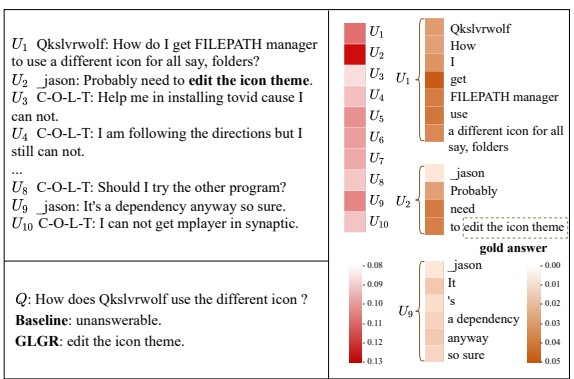

Figure 4: A case study for the ELECTRA-based GLGR

subsection, we intend to compare them to the global-to-local two-stage GLGR.

For the single-phase reasoning version, we combine QIUG with LSRG by two steps, including 1) connecting noun phrases in the question node to the similar fine-grained nodes in utterances, and 2) connecting utterance nodes to the entities occurring in them. On this basis, we encode the resultant holistic graph using GAT, which serves as the single-phase GLGR. It is equipped with ELECTRA and the pointer network.

The comparison results (single-phase GLGR versus two two-stage GLGRs) are shown in Table 5, where ELECTRA is used as the backbone. It is evident that the single-stage GLGR obtains inferior performance. It is most probably because that the perception of co-attentive information among fine-grained nodes in LSRG suffers from the interference of irrelevant coarse-grained nodes in QIUG. This drawback raises a problem of combining heterogeneous graphs for multi-hop reasoning. Besides, we observe that the global-to-local reasoning method exhibits better performance compared to local-to-global graph reasoning. We attribute it to the initial local graph reasoning in the local-to-global reasoning, which ineffectively integrates the distant and important context information while focusing on local semantic information. This leads to suboptimal multi-hop reasoning and highlights the importance of the graph reasoning order in handling complex information dependencies.

## 3.8 Case Study

GLGR is characterized as the exploration of both global and local clues for reasoning. It is implemented by highlighting co-attentive information in coarse-grained and fine-grained nodes.

Figure 4 shows a case study for GLGR-based MDRC, where the heat maps exhibit the attention distribution on both utterance nodes and token-level nodes. There are three noteworthy phenomena occurring in the heat maps. First, GLGR assigns higher attention scores to two utterance nodes, which contain the answer and closely-related clues, respectively. Second, both the answer and clues are assigned higher attention scores, compared to other token-level nodes. Finally, the answer and clues emerge from different utterance nodes.

This is not an isolated case, and the phenomena stand for the crucial impacts of GLGR on MDRC.

## 4 Related Work

### 4.1 Multi-party Dialogue Reading Comprehension

A variety of graph-based approaches have been studied for MDRC. They successfully incorporate conversational structural features into the dialogue modeling process.

Ma et al. (2021) construct a provenance-aware graph to enhance the co-attention encoding of discontinuous utterances of the same interlocutor. Li et al. (2021) and Ma et al. (2022) apply CDS to bring the mutual dependency features of utterances into the graph reasoning process, where GCN (Kipf and Welling, 2017) is used for encoding. Recently, Li et al. (2022) propose a back-and-forth comprehension strategy. It decouples the past and future dialogues, and models interactive relations in terms of conversational temporality. Li et al. (2023) add the coreference-aware attention modeling in PLM to strengthen the multi-hop reasoning ability.

Another research branch focuses on the study of language understanding for dialogues, where self-supervised learning is used for the general-to-specific modeling and transfer of the pretrained models. Li and Choi (2020) transfer BERT to the task-specific data, where two self-supervised tasks are used for fine-tuning, including Utterance-level Masked Language Modeling (ULM) and Utterance-Order Prediction (UOP). During the transfer, the larger-sized dataset of FriendsQA transcripts is used. Similarly, Li and Zhao (2021) transfer PLMs to dialogues using a multi-task learning framework,

including the tasks of interlocutor prediction and key utterance prediction.

## 4.2 Semantic Role Labeling

In this study, we use Semantic Role Labeling (SRL) to build LSRG for local graph reasoning. To facilitate the understanding of SRL, we present the related work as follows.

SRL is a shallow semantic parsing task that aims to recognize the predicate-argument structure of each sentence. Recently, advances in SRL have been largely driven by the development of neural networks, especially the Pre-trained Language Models (PLMs) such as BERT (Kenton and Toutanova, 2019). Shi and Lin (2019) propose a BERT-based model that incorporates syntactic information for SRL. Larionov et al. (2019) design the first pipeline model for SRL of Russian texts.

It has been proven that SRL is beneficial for MRC by providing rich semantic information for answer understanding and matching. Zheng and Kordjamshidi (2020) introduce an SRL-based graph reasoning network to the task of multi-hop question answering. They demonstrate that the fine-grained semantics of an SRL graph contribute to the discovery of an interpretable reasoning path for answer prediction.

## 5 Conclusion

We propose a global-to-local graph reasoning approach towards MDRC. It explores attentive clues for reasoning in both coarse-grained graph QIUG and fine-grained graph LSRG. Experimental results show that the proposed approach outperforms the PLMs baselines and state-of-the-art models. The codes are available at `https://github.com/YanLingLi-AI/GLGR`.

The main contribution of this study is to jointly use global conversational structures and local semantic structures during encoding. However, it can be only implemented by two-stage reasoning due to the bottleneck of in-coordinate interaction between heterogeneous graphs. To overcome the issue, we will use the pretrained Heterogeneous Graph Transformers (HGT) for encoding in the future. Besides, the graph-structure based pretraining tasks will be designed for task-specific transfer learning.

## Limitations

While GLGR demonstrates several strengths, it also has limitations that should be considered. First, GLGR relies on annotated conversation structures, co-reference, and SRL information. This dependency necessitates a complex data preprocessing process and makes the model susceptible to the quality and accuracy of the annotations. Therefore, it is important to ensure the accuracy and robustness of the annotations used in the model training and evaluation process. Second, GLGR may encounter challenges in handling longer dialogue contexts. The performance may exhibit instability when confronted with extended and more intricate conversations. Addressing this limitation requires further investigation of the stability and consistency in a real application scenario.

## Acknowledgements

The research is supported by National Key R&D Program of China (2020YFB1313601), National Science Foundation of China (62376182, 62076174) and Institute of Infocomm Research of A*STAR (CR-2021-001).

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
