# OpenReview forum: "GLGR: Question-aware Global-to-Local Graph Reasoning for Multi-party Dialogue Reading Comprehension"
_EMNLP/2023/Conference — EMNLP 2023 Findings_

### Official Review · Reviewer_BDx5 · 2023-08-02

**Soundness:** 4

**Excitement:**

3: Ambivalent: It has merits (e.g., it reports state-of-the-art results, the idea is nice), but there are key weaknesses (e.g., it describes incremental work), and it can significantly benefit from another round of revision. However, I won't object to accepting it if my co-reviewers champion it.

**Paper Topic And Main Contributions:**

The paper, "GLGR: Question-aware Global-to-Local Graph Reasoning for Multi-party Dialogue Reading Comprehension (MDRC)," addresses the problem of multi-hop reasoning over global conversational structures and the omission of token-level semantic relations in utterances. The authors propose a Global-to-Local Graph Reasoning approach (GLGR) with pre-trained language models to encode Question-aware Interlocutor-Utterance Graph (QIUG) and Local Semantic Role Graph (LSRG). The primary contributions of the paper are the proposition of a question-aware global-to-local graph reasoning approach, the implementation of this approach using a two-stage encoder network, and an experimental validation of this approach using benchmark datasets.

**Questions For The Authors:**

A: Will the proposed method have a higher computational cost compared to other baselines in the paper?

**Reasons To Accept:**

- This paper proposes an innovative reasoning approach, GLGR, tackling the problem of MDRC that has not been well-addressed.
- The paper makes a decent contribution by modeling the Question-aware Interlocutor-Utterance Graph (QIUG) and Local Semantic Role Graph (LSRG) and demonstrating improvements in MDRC performance compared to PLM-based baselines.
- The solution proposed in this paper displays versatility as it shows general improvements regardless of the PLM adopted as the backbone.

**Reasons To Reject:**

- The adoption of a two-stage encoder network for implementing the reasoning process may be computationally expensive and potentially limit its feasibility in large-scale applications.

**Reproducibility:**

4: Could mostly reproduce the results, but there may be some variation because of sample variance or minor variations in their interpretation of the protocol or method.

**Reviewer Confidence:**

2: Willing to defend my evaluation, but it is fairly likely that I missed some details, didn't understand some central points, or can't be sure about the novelty of the work.

---

> ### Author Rebuttal · Authors · 2023-08-27
>
> We wholeheartedly appreciate your dedicated time in reviewing our paper and offering valuable suggestions to enhance its quality. Your feedback holds immense value for us, and we approach it with utmost seriousness. Each suggestion you provided has been thoroughly considered, and we provide our responses as follows:
>
> ----------Response to Comments----------
>
> **Comment #1:** The adoption of a two-stage encoder network for implementing the reasoning process may be computationally expensive and potentially limit its feasibility in large-scale applications.
>
> **Response to #1:** Our computations are predominantly centered around graph modeling, where both QIUG and LSRG incorporate two layers of graph neural networks. Specifically, if we utilize the ELECTRA backbone (335 million parameters), GLGR’s parameter count would only rise by around 45 million. Furthermore, the backbone we utilize is notably more lightweight compared to LLM such as Llama (7 billion parameters). Our models were trained and tested in a mid-range server that possesses a V100 GPU.
>
>
> ----------Response to Questions----------
>
> **Question #1:** Will the proposed method have a higher computational cost compared to other baselines in the paper?
>
> **Response to #1:** The computations in GLGR primarily revolve around the graph reasoning modules QIUG and LSRG. Each module has two layers of graph neural networks. Consequently, the increase in parameter count remains manageable. If we use the ELECTRA backbone (335 million parameters), the parameter count would rise by 45 million. The overview of parameter counts of the previous work is as follows.
>
> SKIDB--- It employs a multi-task learning framework with speaker and key utterance prediction tasks. More parameters come from speaker modeling and utterance prediction. Specifically, SKIDB employs 3 to 5 transformer blocks for speaker information flow and extra layers for utterance prediction. Compared to SKIDB, GLGR's parameter count increase is modest.
>
> ESA--- With speaker masking, speaker graph and discourse graph, ESA adds three channels to the backbone using R-GCN for graph modeling. By contrast, GLGR uses two graphs (QIUG and LSRG), possibly resulting in a parameter count similar to or lower than ESA.
>
> BiDeN--- It uses three channels to capture latent temporal information in the dialogue, including past-to-present, present-to-present and future-to-present channels. Each channel consists of multiple transformer blocks. The majority of additional parameters in BiDeN come from these transformer blocks. Considering that the parameter count for a single layer of a graph neural network is comparable to that of a transformer block, the incremental parameter increase in our GLGR is not substantial, compared to BiDeN.
>
> ULM-UOP--- It utilizes BERT (12 transformer encoder blocks). Due to additional pretraining, it consumes more resources during training than GLGR.
>
> DADgraph--- It uses BERT without fine-tuning, relying solely on its embeddings. It employs a global utterance graph for discourse and speaker structure, resulting in lower training computational demand compared to GLGR. However, GLGR obtained considerable performance enhancements compared to DADgraph.
>
> Therefore, the computational cost of GLGR is insignificantly higher than most of the previous work.
>
>
> Thank you again for your insightful feedback, which undoubtedly contributes to enhancing the quality and enriching the final version of the paper.

---

### Official Review · Reviewer_wxqV · 2023-08-03

**Soundness:** 3

**Excitement:**

3: Ambivalent: It has merits (e.g., it reports state-of-the-art results, the idea is nice), but there are key weaknesses (e.g., it describes incremental work), and it can significantly benefit from another round of revision. However, I won't object to accepting it if my co-reviewers champion it.

**Paper Topic And Main Contributions:**

This  manuscript proposes a question-aware global-to-local graph reasoning approach GLGR by integrating QIUG and LSRG. Compared to PLM-based baselines, GLGR achieves better results.

**Reasons To Accept:**

1. Manuscript  is well organized and written.

2. GLGR integrates global reasoning and local graph reasoning methods, and achieves substantial improvements.

3. The authors did some meaningful integration work.

**Reasons To Reject:**

1. The method of comparison is too old-fashioned and the latest baseline BiDEN is not for MDRC(Multi-party Dialogue Reading Comprehension).

2. Insufficient task-related baseline comparisons.

3. The method of construction graph used by the authors seems to be a relatively common technique used in neural networks.

4. In line 292~294, why is the threshold set to 0.5? Are $W^l$ (in Equation 3) and $W^l_t $ in line 251 the same?

5. According to the parameter settings, L1 is set to 2. But the two layers graph neural network cannot obtain global information, it can only obtain the node information of the second hop. Does the author need to discuss the impact of the number of graph neural network layers on model performance? Does global information help the model?

**Reproducibility:**

4: Could mostly reproduce the results, but there may be some variation because of sample variance or minor variations in their interpretation of the protocol or method.

**Reviewer Confidence:**

4: Quite sure. I tried to check the important points carefully. It's unlikely, though conceivable, that I missed something that should affect my ratings.

---

> ### Author Rebuttal · Authors · 2023-08-27
>
> We extend our sincere appreciation for your dedicated effort in reviewing our paper and offering valuable suggestions to enhance its quality. Your feedback holds significant value for us, and we genuinely consider each suggestion you provide. We take your feedback seriously, and our response is as follows:
>
>
> ----------Response to Comments----------
>
> **Comment #1:** The method of comparison is too old-fashioned and the latest baseline BiDEN is not for MDRC(Multi-party Dialogue Reading Comprehension).
>
> **Response to #1:** BiDEN is a model designed for dialog comprehension. It has been extensively tested using Molweni dataset, although the experiment isn’t conducted by us. More details can be found in Li et al (EMNLP, 2022) [1]’s work which provides the experimental results. Considering that Multi-party Dialogue Reading Comprehension (MDRC) can be regarded as a subtask of dialog comprehension, we kindly suggest that the utilization of BiDEN for MDRC is feasible.
>
>
> **Comment #2:** Insufficient task-related baseline comparisons.
>
> **Response to #2:** Within Table 2, an extensive comparative analysis is performed, which involves three baseline models, including BERT-base, BERT-large and ELECTRA. These models constitute all the pre-trained language models employed in the previous work. We systematically compare and analyze these baseline models, and reveal that GLGR exhibits noticeable performance enhancements. Furthermore, we have conducted comprehensive comparisons with all the task-related approaches that were published before the deadline of EMNLP 2023.
>
>
> **Comment #3:** The method of construction graph used by the authors seems to be a relatively common technique used in neural networks.
>
> **Response to #3:** Graph-based neural networks are commonly employed to address comprehension tasks, the ones that involve intricate interactions among various forms of structured information. However, the effectiveness of these methods can be contingent on the task and dataset at hand. The direct transfer of existing graph-based models often encounters limitations due to the distinctive characteristics of different tasks or datasets. In MDRC, the utilization of global utterance graphs is a conventional approach. Nevertheless, when faced with the challenge of accurately capturing local semantic interactions, we take steps to model localized semantic graphs. More importantly, we introduce a novel approach by combining global and local graphs, incorporating a multi-level graph reasoning methodology that progresses from coarse to fine-grained reasoning. In the MDRC task, our contribution lies in the novel design of graph construction and modeling at the level of dialogue structure. To our best knowledge, there are no other studies that have adopted similar strategies.
>
>
> **Comment #4:** In line 292~294, why is the threshold set to 0.5? Are $W^{l}$ (in Equation 3) and $W_t^{l}$ in line 251 the same?
>
> **Response to #4:** The setting of the threshold (0.5) at Line 291 is a result of empirical observation. This value is established after manually assessing around 100 graph nodes. For example, there is an overlap rate greater than 0.5 between “there is any way to disable tapping from my touchpad” and “disable tapping from my touchpad”, whereas the overlap rate between “it's really annoying” and “disable tapping from my touchpad” is less than 0.5. Regarding the differentiation between the two trainable weights $W^{l}$ and $W_t^{l}$, the latter is a specialized description of the former. $W_t^{l}$ is a specific weight of the t-th edge.
>
>
> **Comment #5:** According to the parameter settings, L1 is set to 2. But the two-layer graph neural network cannot obtain global information, it can only obtain the node information of the second hop. Does the author need to discuss the impact of the number of graph neural network layers on model performance? Does global information help the model?
>
> **Response to #5:** We kindly suggest that the comment is about the configuration of neural network layers of QIUG and LSRG. The experimental results demonstrate that a two-layer architecture (L1=2) outperforms a single-layer structure. However, extending the structure with more layers (beyond 2) doesn’t yield substantial performance improvements. We consider this phenomenon to be attributed to the characteristics of Molweni and FriendsQA datasets. In these datasets, the majority of questions entail reasoning steps that do not surpass two iterations. Furthermore, we kindly suggest that introducing multiple layers into the graph neural network will significantly increase the number of parameters. Considering the delicate equilibrium between parameter count and performance, we designate both L1 and L2 as 2 layers.
>
>
> ----------Reference----------
>
> [1] Yiyang Li, Hai Zhao and Zhousheng Zhang. Back to the future: Bidirectional information decoupling network for multi-turn dialogue modeling. In EMNLP 2022.
>
> We sincerely appreciate the constructive critique provided by the reviewers, which has undeniably enriched the final version of the paper. Thank you for your time and consideration.

---

### Official Review · Reviewer_28uk · 2023-08-09

**Soundness:** 3

**Excitement:**

3: Ambivalent: It has merits (e.g., it reports state-of-the-art results, the idea is nice), but there are key weaknesses (e.g., it describes incremental work), and it can significantly benefit from another round of revision. However, I won't object to accepting it if my co-reviewers champion it.

**Paper Topic And Main Contributions:**

* The paper proposes a question-aware global-to-local graph reasoning approach (GLGR) for multi-party dialogue reading comprehension (MDRC).

* The approach encodes two heterogeneous graphs - a Question-aware Interlocutor-Utterance Graph (QIUG) that captures global interactive relations, and a Local Semantic Role Graph (LSRG) that captures local semantic relations within utterances. By developing a two-stage encoder network to implement progressive graph reasoning from global QIUG to local LSRG, this technique allows integrating both global and local clues for answer prediction.

* By running experiments on two MDRC datasets (FriendsQA and Molweni) show that GLGR achieves significant improvements over BERT and ELECTRA baselines and state-of-the-art methods. The results show the contributions of global and local graph reasoning, as well as their integration via the two-stage reasoning process.

**Questions For The Authors:**

Please address the concerns in reasons to reject.

**Reasons To Accept:**

* The two-stage reasoning framework going from global graph to local graph is intuitive and well-motivated for connecting coarse-grained and fine-grained semantics.
* Good empirical performance improvements over baselines.
* Generalization across models with different pretraining schemes (BERT and ELECTRA) which is promising

**Reasons To Reject:**

* It would be good to see some additional qualitative analysis of the outputs.
* The decrease in performance as context length increases is a concern for practical applicability of these models. Have any experiments been tried with enc-dec models (ex. FlanT5) or decoder only models that have been pretrained with longer context lengths? Would that solve the issue?
* The technique proposed seems to include a lot of moving parts and engineering effort compared to the performance improvement over other baselines considered, which again impacts practical applicability. I would like to hear the authors' opinion on this.

**Reproducibility:**

3: Could reproduce the results with some difficulty. The settings of parameters are underspecified or subjectively determined; the training/evaluation data are not widely available.

**Reviewer Confidence:**

3: Pretty sure, but there's a chance I missed something. Although I have a good feel for this area in general, I did not carefully check the paper's details, e.g., the math, experimental design, or novelty.

---

> ### Author Rebuttal · Authors · 2023-08-27
>
> We would like to express our heartfelt appreciation for dedicating your valuable time to reviewing our paper and offering us invaluable insights and suggestions. Your feedback is of immense significance to us and has played an important role in refining our work. We have taken careful consideration of each of your comments, and we provide our responses to each point as below.
>
> ----------Response to Comments----------
>
> **Comment #1:** It would be good to see some additional qualitative analysis of the outputs.
>
> **Response to #1:** Our interpretation of qualitative analysis is to understand how GLGR influences the behavior of the baseline models. In the case study mentioned in Line 569 of the paper, we present a detailed instance along with corresponding heatmaps. They effectively visualize the impact of using GLGR on answer extraction. If this style of qualitative analysis aligns with your requirements, we can provide additional instances for demonstration as long as the length of the paper is allowed. Furthermore, we can also add an appendix to dissect scenarios where the baseline models fall short and elucidate how GLGR enhances performance in those aspects.
>
>
> **Comment #2:** The decrease in performance as context length increases is a concern for practical applicability of these models. Have any experiments been tried with enc-dec models (e.g. FlanT5) or decoder-only models that have been pretrained with longer context lengths? Would that solve the issue?
>
> **Response to #2:** It is interesting to utilize encoder-decoder models like FlanT5 or decoder-only models within a generative framework. However, it is important to note that FlanT5 often comes with larger parameter counts, which could potentially pose limitations in scenarios with limited resources. More importantly, the datasets we used, i.e., Molweni and FriendsQA, are designed specifically for extractive machine reading comprehension. FlanT5 is developed within a generative framework. It is difficult to fairly evaluate the performance of FlanT5 using Molweni and FriendsQA. It is because the proper answers generated by FlanT5 may be determined as incorrect when they don’t occur in the dialogues. By contrast, BERT and ELECTRA are constructed with transformer encoders, and they are widely used in the extractive framework in this field. Therefore, we present the performance of our models that use BERT and ELECTRA as backbones.
>
> **Comment #3:** The technique proposed seems to include a lot of moving parts and engineering effort compared to the performance improvement over other baselines considered, which again impacts practical applicability. I would like to hear the authors' opinion on this.
>
> **Response to #3:**  We consider that this concern pertains to the necessity of employing NLP models to extract coreference and semantic role structures from the dialogues before reasoning. Detailed information on this aspect can be found in Lines 276 and 296. Specifically, the coreference resolution model we employ attains the following performance on the Conll-2012 test set, including the precision of 0.79, recall of 0.78 and F1-score of 0.79. More details can be found by accessing the hyperlink of “https://demo.allennlp.org/coreference-resolution”. In addition, the semantic role labeling model achieves an F1-score of 0.86 on the Ontonotes 5.0 test set. More details can be found by accessing the hyperlink of “https://demo.allennlp.org/semantic-role-labeling”. The above performance demonstrates that there's potential for improvement in the performance of these models. Furthermore, the models have been designed for general applicability and have not been specifically trained for multi-party dialogue contexts. Nevertheless, the coreference and semantic role label information obtained by these techniques contribute to reasoning in the MDRC task. As demonstrated in Table 4, incorporating both coreference and semantic role information leads to improvement. Briefly, we kindly suggest that the utilization of general information extraction techniques endows GLGR with a degree of adaptability in practical scenarios.
>
> We extend our heartfelt gratitude for your insightful comments, which will help us improve the quality of our paper.

---

### Official Review · Reviewer_F64H · 2023-08-10

**Soundness:** 4

**Excitement:**

3: Ambivalent: It has merits (e.g., it reports state-of-the-art results, the idea is nice), but there are key weaknesses (e.g., it describes incremental work), and it can significantly benefit from another round of revision. However, I won't object to accepting it if my co-reviewers champion it.

**Paper Topic And Main Contributions:**

The paper introduces two types of graphs to facilitate reasoning in the context of multi-party dialogue reading comprehension. These graphs model global information across utterances and local information within individual utterances. A two-stage encoder network is employed to reason from the global level to the local level. Experimental results demonstrate that the global-to-local reasoning approach using the two types of graphs leads to significant improvements compared to baseline models.

**Questions For The Authors:**

A: Have you conducted an ablation study on Question-aware Reasoning (line 341) to verify its effectiveness?

B: Have you considered reversing the order of graph encoding, i.e., local-to-global? If so, how would it potentially impact the results?

C: Given that longer dialogues can introduce more complex relationships between utterances and entities, have you explored any denoising methods to address this issue?

**Reasons To Accept:**

The idea of constructing multi-grained graphs for reasoning is straightforward yet effective.

The paper proposes an approach to construct Question-aware Interlocutor-Utterance Graph and Local Semantic Role Graph, which are crucial for understanding the underlying semantic structure necessary for reasoning.

The global-to-local reasoning method applied to both graphs results in improved performance in multi-party dialogue reading comprehension.

The paper includes detailed experiments and an ablation study to demonstrate the effectiveness of the proposed approach.

**Reasons To Reject:**

The proposed graph construction heavily relies on external tools, such as co-reference, which may impact the stability of the approach.

As dialogues become longer, the relationships between utterances and entities can become complex and introduce additional noise, thereby limiting the applicability and generalization ability of the approach.

The effect of Question-aware Reasoning (line 341) has not been sufficiently validated.

**Reproducibility:**

3: Could reproduce the results with some difficulty. The settings of parameters are underspecified or subjectively determined; the training/evaluation data are not widely available.

**Reviewer Confidence:**

4: Quite sure. I tried to check the important points carefully. It's unlikely, though conceivable, that I missed something that should affect my ratings.

---

> ### Author Rebuttal · Authors · 2023-08-27
>
> We extend our sincere gratitude for dedicating your valuable time to reviewing our paper and providing us with invaluable insights and suggestions. Your feedback is greatly appreciated and has undoubtedly contributed to the refinement of our work. We have meticulously considered each of your suggestions and comments. We answer the concerns as follows:
>
> ----------Response to Comments----------
>
> **Comment #1:** The proposed graph construction heavily relies on external tools, such as co-reference, which may impact the stability of the approach.
>
> **Response to #1:** We acknowledge your concern regarding the dependence of the LSRG's construction on other NLP technologies. These technologies are mentioned in Lines 276 and 296 of the paper. Let us briefly introduce their performance first. The performance of the coreference resolution toolkit on the Conll-2012 test set stands at the precision of 0.79, recall of 0.78 and F1-score of 0.79. Detailed information can be found by accessing the hyperlink of “https://demo.allennlp.org/coreference-resolution”. The semantic role label toolkit achieves an F1-score of 0.86 on the Ontonotes-5.0 test set. Detailed information can be found by accessing the hyperlink of “https://demo.allennlp.org/semantic-role-labeling”. It can be observed that, the performance of these techniques is not flawless. Furthermore, these models are not explicitly tailored to accommodate multi-party dialogue contexts. They are intended for broader domains. Nevertheless, the extracted co-reference and semantic role information remain effective within our approach, as evidenced in Table 4 (between Lines 485-486). The incorporation of both coreference and semantic-role structural information demonstrably enhances model performance. Therefore, the co-reference and semantic role information are pragmatically applicable to our task in terms of model stability, and their influence on the GLGR’s stability remains well-manageable.
>
>
> **Comment #2:** As dialogues become longer, the relationships between utterances and entities can become complex and introduce additional noise, thereby limiting the applicability and generalization ability of the approach.
>
> **Response to #2:** We acknowledge your observation regarding the escalating complexity of relationships in longer dialogues, which can indeed impact model performance. This trend is also illustrated in Figure 3, and it is explained in Lines 517-518. Nevertheless, it is worth highlighting that Figure 3 also reveals a compelling trend, i.e., GLGR effectively mitigates performance decline compared to the baseline models as dialogues are lengthened. In particular, in conversations with over 30 utterances, GLGR's performance remains more stable compared to the baseline models (i.e., the baseline models experience more significant degradation). This demonstrates the relative stability of GLGR within extensive conversational contexts, and suggests its potential for enhancing adaptability in real-world application scenarios.
>
>
> **Comment #3:** The effect of Question-aware Reasoning (line 341) has not been sufficiently validated.
>
> **Response to #3:** The opinion is highly valuable. Our exploration of Question-aware Reasoning primarily centers on QIUG, as we consider that it holds significant pertinence within the QIUG reasoning. Our ablation experiments are outlined in Table 4, spanning Lines 485-486 of the paper. To further delve into the analysis of Question-aware Reasoning within LSRG (as indicated in Line 341), we plan to introduce an additional ablation experiment in subsequent sections. This analysis aims to provide a thorough understanding of the impact of this aspect on GLGR's performance.
>
>
> ----------Response to Questions----------
>
> **Question #1:** Have you conducted an ablation study on Question-aware Reasoning (line 341) to verify its effectiveness?
>
> **Response to #1:** Our investigation into Question-aware Reasoning currently centers on the QIUG reasoning component, supported by specific ablation experiments detailed in Table 4 (Lines 485-486). Concerning the effect of Question-aware Reasoning in LSRG (Line 341), we plan to include this ablation experiment subsequently.
>
>
> **Question #2:** Have you considered reversing the order of graph encoding, i.e., local-to-global? If so, how would it potentially impact the results?
>
> **Response to #2:** The prevalent trend of dialogue modeling focuses largely on the global aspect. In order to strengthen the models with local semantic and interaction features, we formulated the notion of incremental refinement in reasoning. GLGR conducts discourse reasoning at the global level to facilitate the comprehension of structural utterance information. Subsequently, semantic-level local graph inference is employed to optimize comprehension of crucial semantics. Such an incremental refinement process is based on the hypothesis that global-to-local mode exhibits greater explicability compared to local-to-global. However, the local-to-global graph reasoning approach is interesting, and we intend to experiment with it in subsequent phases.
>
>
> **Question #3:** Given that longer dialogues can introduce more complex relationships between utterances and entities, have you explored any denoising methods to address this issue?
>
> **Response #3:** We appreciate your observation regarding the potential complexities arising from extended dialogues. Our interpretation of the denoising approach revolves around the endeavor to meticulously distinguish linguistic structures within multi-party conversations. This serves to mitigate potential disturbances caused by extraneous discourse. The GLGR model employs a global-to-local strategy, i.e., commencing with global-level utterance reasoning and subsequently delving into semantic comprehension of localized utterances. It leverages question-related cues to discern critical linguistic elements associated with the question. In contrast to a comprehensive global graph reasoning conducted all at once, GLGR adopts a more refined approach, intensifying the perception of the interaction between the question and the dialogue. Essentially, this process embodies a coarse-to-fine denoising progression.
>
>
> We are grateful for your insightful comments, which undoubtedly strengthen the quality of our paper. Your guidance has been invaluable in shaping our work into a more coherent and impactful work.

---

### Meta-Review · Area_Chair_mpB3 · 2023-09-19

**Recommendation:** 3

**Metareview:**

Paper Topic And Main Contributions:
* The paper proposes a question-aware global-to-local graph reasoning approach (GLGR) for multi-party dialogue reading comprehension (MDRC).
* It introduces two types of graphs to facilitate reasoning. These graphs model global information across utterances and local information within individual utterances. A two-stage encoder network is employed to reason from the global level to the local level.
* Experiments on two MDRC datasets (FriendsQA and Molweni) show that GLGR achieves significant improvements over BERT and ELECTRA baselines and state-of-the-art methods. The results show the contributions of global and local graph reasoning, as well as their integration via the two-stage reasoning process.

Reasons to accept:
* The paper is well organized and written.
* The two-stage reasoning framework going from global graph to local graph is intuitive and well-motivated for connecting coarse-grained and fine-grained semantics.
* The paper includes detailed experiments and an ablation study to demonstrate the effectiveness of the proposed approach.
* Generalization across models with different pretraining schemes (BERT and ELECTRA), which is promising.
* The global-to-local reasoning method applied to both graphs results in improved performance in multi-party dialogue reading comprehension.

Reasons to reject:
* The proposed graph construction heavily relies on external tools, such as co-reference, which may impact the stability of the approach. The technique proposed seems to include a lot of moving parts and engineering effort compared to the performance improvement over other baselines considered, which again impacts practical applicability. The authors explain in the rebuttal that their models are more lightweight than LLMs such as Llama.
* The decrease in performance as context length increases is a concern for practical applicability of these models. As dialogues become longer, the relationships between utterances and entities can become complex and introduce additional noise, thereby limiting the applicability and generalization ability of the approach. The authors explain in the rebuttal that their model performs better than the baselines in this regard.

---

### Decision · Program_Chairs · 2023-10-07

**Decision:**

Accept-Findings

**Comment:**

Paper Topic And Main Contributions:
* The paper proposes a question-aware global-to-local graph reasoning approach (GLGR) for multi-party dialogue reading comprehension (MDRC).
* It introduces two types of graphs to facilitate reasoning. These graphs model global information across utterances and local information within individual utterances. A two-stage encoder network is employed to reason from the global level to the local level.
* Experiments on two MDRC datasets (FriendsQA and Molweni) show that GLGR achieves significant improvements over BERT and ELECTRA baselines and state-of-the-art methods. The results show the contributions of global and local graph reasoning, as well as their integration via the two-stage reasoning process.

Reasons to accept:
* The paper is well organized and written.
* The two-stage reasoning framework going from global graph to local graph is intuitive and well-motivated for connecting coarse-grained and fine-grained semantics.
* The paper includes detailed experiments and an ablation study to demonstrate the effectiveness of the proposed approach.
* Generalization across models with different pretraining schemes (BERT and ELECTRA), which is promising.
* The global-to-local reasoning method applied to both graphs results in improved performance in multi-party dialogue reading comprehension.

Reasons to reject:
* The proposed graph construction heavily relies on external tools, such as co-reference, which may impact the stability of the approach. The technique proposed seems to include a lot of moving parts and engineering effort compared to the performance improvement over other baselines considered, which again impacts practical applicability. The authors explain in the rebuttal that their models are more lightweight than LLMs such as Llama.
* The decrease in performance as context length increases is a concern for practical applicability of these models. As dialogues become longer, the relationships between utterances and entities can become complex and introduce additional noise, thereby limiting the applicability and generalization ability of the approach. The authors explain in the rebuttal that their model performs better than the baselines in this regard.